# An Open Soil Structure Library based on X-ray CT data

Ulrich Weller[1], Lukas Albrecht[2], Steffen Schlüter[1], and Hans-Jörg Vogel[1,3]

[1]Department of Soil System Science, Helmholtz Centre for Environmental Research - UFZ, Theodor-Lieser-Str. 4, 06120 Halle (Saale), Germany
[2]Agroscope, Agroecology and Environment, Soil Quality and Soil Use, Reckenholzstrasse 191, 8046 Zurich, Switzerland
[3]Martin-Luther-University Halle-Wittenberg, Institute of Soil Science and Plant Nutrition, Von-Seckendorff-Platz 3, 06120 Halle (Saale), Germany

**Correspondence:** Ulrich Weller (ulrich.weller@ufz.de)

**Abstract.** Soil structure in terms of the spatial arrangement of pores and solid is highly relevant for most physical and biochemical processes in soil. While this is known for long, a scientific approach to quantify soil structural characteristics was also missing for long. This was due to its buried nature but also due to the three-dimensional complexity.

During the last two decades, tools to acquire full 3D images of undisturbed soil became more and more available and a number of powerful software tools were developed to reduce the complexity to a set of meaningful numbers. However, the standardization of soil structure analysis for a better comparability of the results is not well developed and the accessibility of required computing facilities and software is still limited. At this stage we introduce an open access *Soil Structure Library* (https://structurelib.ufz.de/) which offers well-defined soil structure analyses for X-ray CT data sets uploaded by interested scientists. At the same time, the aim of this library is to serve as an open data source for real pore structures as developed in a wide spectrum of different soil types under different site conditions all over the globe, by making accessible the uploaded binarised 3D images. By combining pore structure metrics with essential soil information requested during upload (e.g. bulk density, texture, organic carbon content. . . ), this Soil Structure Library can be harnessed towards data mining and development of soil structure based pedotransfer functions.

In this paper we describe the architecture of the *Soil Structure Library* and the provided metrics. This is complemented by an example how the data base can be used to address new research questions.

## 1 Introduction

Soil structure is of central importance for soil functions. Besides its relevance for plant growth, this is also true for the storage and movement of water and solutes inside the soil pore system, for biochemical matter cycling and for soil as habitat for a myriad of interacting organisms (Dexter, 1988; Rabot et al., 2018).

For a long time, a crucial hurdle in exploring soil structure was that soil is opaque so that soil structural properties were hardly accessible. This was especially true with respect to quantitative analysis as required for any scientific evaluation.

During the last three decades, with the development and increasing availability of X-ray CT scanners we are now in the position to quantify soil structure without disturbance in full three dimensions and with a spatial resolution of a few microns or

even below. This boosted an enormous amount of scientific insight especially with respect to the soil pore structure in relation to water dynamics and solute transport (Wildenschild and Sheppard, 2013; Larsbo et al., 2014; Tracy et al., 2015). More recently also the importance of soil structure for the turnover of organic matter (Kravchenko et al., 2019; Schlüter et al., 2022b) and as habitat for soil organisms (Juarez et al., 2013; Falconer et al., 2012; Juyal et al., 2019) is studied based on 3D images.

In parallel, software tools especially tailored for the analysis of 3D porous media analysis were developed, partly as commercial products (MAVI) partly as open source applications and program libraries such as QuantIm (www.quantim.ufz.de), or ImageJ/FIJI (Schindelin et al., 2012) with dedicated 3D structure analysis plugins like SoilJ (Koestel, 2018), BoneJ (Doube et al., 2010), MorpholibJ (Legland et al., 2016) and the python software sci-kit-image (van der Walt et al., 2014).

Fortunately, the available computing power increased together with the size of the images generated by X-ray CT scanners. However, when it comes to the calculation of distance distributions and connectivity measures, a considerable amount of computing power is required which often exceeds the capacity of standard computers. Another difficulty is the lack of comparability of the results, since the detailed algorithms to calculate soil structural attributes such as connectivity or pore size distribution are not always obvious. Hence some standardised analysis would be beneficial to generate results that are comparable among different studies.

The motivation of the *Soil Structure Library* (https://structurelib.ufz.de/) introduced in this paper is to offer some standardized analysis of the 3D pore structure obtained from X-ray CT together with the required computing power. The price we charge for this service is that the analyzed structures are made freely available through our web site together with the metadata describing the soil. This should generate some substantial benefit for both, the data providers, who get standardized analysis for their CT images, and for the wider scientific community, who gets access to a wide range of soil structure data including additional information on the specific climate, land use and soil type.

It should be noted that the provided analysis is limited to the analysis of binary images. This means that the user needs to upload images which are already segmented into pore and solid. We are aware that segmentation is a crucial step in image analysis and there are no objective procedures how to do it (Baveye et al., 2010; Schlüter et al., 2014). This is why we prefer to leave this step to the data owner who, however, has to upload at least one 2D image of the original grey scale CT image so that the effect of the segmentation process is illustrated and can be understood by others.

## 2   General description of soil structure library

Our soil structure library is open for anybody with a clear focus on the soil science community. For uploading image data, the user has to subscribe and to ascertain the data policy. For each uploaded data set metadata on soil and site properties are requested. A part of this information is mandatory, another part is optional. Table 1 gives an overview of the metadata.

Although many of the parameters are listed as optional, it is highly recommended to provide a rather complete list, as this information will make the data much more valuable for others. The complete content of the database, including the 3D binarised X-ray images, is provided under Creative Commons and thus can be used for further research by the entire scientific community.

| property | requirement | remarks |
|---|---|---|
| classification system | optional | system used for soil type and texture class |
| soil type | optional | according to classification system |
| location | mandatory | latitude, longitude (WGS84 assumed) and location name |
| land use | mandatory | main land use at sampling date e.g. pasture |
| tillage | optional | main tillage at sampling date e.g. no tillage |
| crop rotation | optional | listed backwards, starting with crop at sampling date |
| sampling date | optional | YYYY-MM-DD format |
| sample height | optional | indicate in cm |
| soil depth | mandatory | depth of the sample top in cm |
| sample type | mandatory | undisturbed/repacked |
| segmetation | optional | definition of segmentation procedure |
| texture | optional | sand, silt and clay content |
| texture class | mandatory | according to classification system |
| voxel size | mandatory | indicate in mm |
| bulk density | optional | indicate in g cm$^{-3}$ |
| organic carbon content | optional | indicate in TOC g kg$^{-1}$ fine earth |
| XY Panel | mandatory | representative plane of binary and grey value image in .tiff format |
| reference | optional | doi that directly links to the corresponding paper |
| comment | optional | additional information e.g. time since last tillage, segmentation method... |

**Table 1.** Summary of meta information

The architecture of the library comprises three servers: a webserver, a database server, and an image processor. The webserver hosts the user frontend, and manages the user administration, data input, file uploads, and the presentation of results and metadata. It is implemented in Django, which integrates data modelling and web service based on Python. Django communicates with the database server configured as a MySql server. The image processing is triggered as soon as the data server receives new data from the frontend. It gets transferred to the image processor consisting of a linux workstation where an ImageJ macro is launched. Upon completion, the results are uploaded to the database server. Simultaneously, the web server sends an email to the submitter to inform about the completion of the calculations.

All machines but the web server are behind a firewall. The only connection across this this firewall is through the database connection. All other required connections are realized behind the wall. The modular structure makes it possible to provide further computing power (i.e. a computing cluster) when needed.

## 2.1 Images Files

Three-dimensional binary images of the pore structure can either be uploaded in the popular 3D tiff format or in the mhd/raw format common in ITK. Zero values will be distinguished from non-zero values, so the actual gray value of the non-zero phase, e.g. 1 or 255, does not matter. It is optional whether zero should be the foreground or background phase (i.e. pore or solid). Also, it is optional to upload a mask to specify a region of interest (ROI) for which the analysis is required. Typical examples would be ROIs for cylindrical soil cores or irregularly shaped soil aggregates. The ROI image can be uploaded in form of a binary image (tiff or mhd/raw) with identical 3D dimensions as the uploaded image, or in form of a selection (roi format) to be created and exported in ImageJ. This selection is a two-dimensional image and will be applied to all slices equally (e.g. a circular selection will result in a cylindrical mask). All files have to be uploaded in one compressed zip-folder. The name of ROI has to be provided upon upload. The remaining image file is considered as the pore structure image.

In addition to the 3D image, two-dimensional slices in at least one but preferably in all three principal directions have to be uploaded both in grayscale and binary form in order to provide information on the quality of image segmentation. Since segmentation has to be done beforehand, it is the only form of quality control warranted in the soil structure library to judge about outliers being caused by natural variation or improper segmentation.

## 2.2 Image Analysis

The segmented image is processed with a standardized workflow implemented as an ImageJ-macro that is executed in the Fiji distribution of ImageJ (Schindelin et al., 2012) and associated plugins. The binary image undergoes several transforms to extract a limited set of meaningful pore metrics (Fig. 1). A complete list of quantitative output with units is summarized in Table 2.

### 2.2.1 Binary pore structure

A first set of structural properties is directly derived from the binary pore structure (Fig. 1a). These Minkowski functionals $M_{0-3}$ (Vogel et al., 2010; Armstrong et al., 2019) comprise fundamental properties of complex objects like volume ($M_0$), surface area ($M_1$), integral of mean curvature ($M_2$), and integral of total curvature ($M_3$). The meaning of $M_0$ and $M_1$ is obvious. $M_2$ is negative for concave surfaces as typical for packing voids in granular media while it is positive for convex surfaces as spherical bubbles or cylindrical pores. For cylindrical pores $M_2$ can be directly related to the length of these pores (Koebernick et al., 2014).

$M_3$ is directly related to the Euler characteristic $\chi$ (Vogel et al., 2010; Armstrong et al., 2019),

$$\chi = M_3/4\pi = \mathcal{N} - \mathcal{L} + \mathcal{O}, \tag{1}$$

a topological number that sums over all isolated objects $\mathcal{N}$ and fully enclosed cavities $\mathcal{O}$ and subtracts the number of redundant loops $\mathcal{L}$. $\mathcal{O}$ is typically negligible, as it represents the number of floating particles in the pore space. Therefore, $\chi$ can be

**Figure 1.** Schematic of a series image transforms and the structure metrics derived from it. The pore structure is taken from a $30 \times 30 \times 10$ mm volume of a fine-textured topsoil managed as no-till.

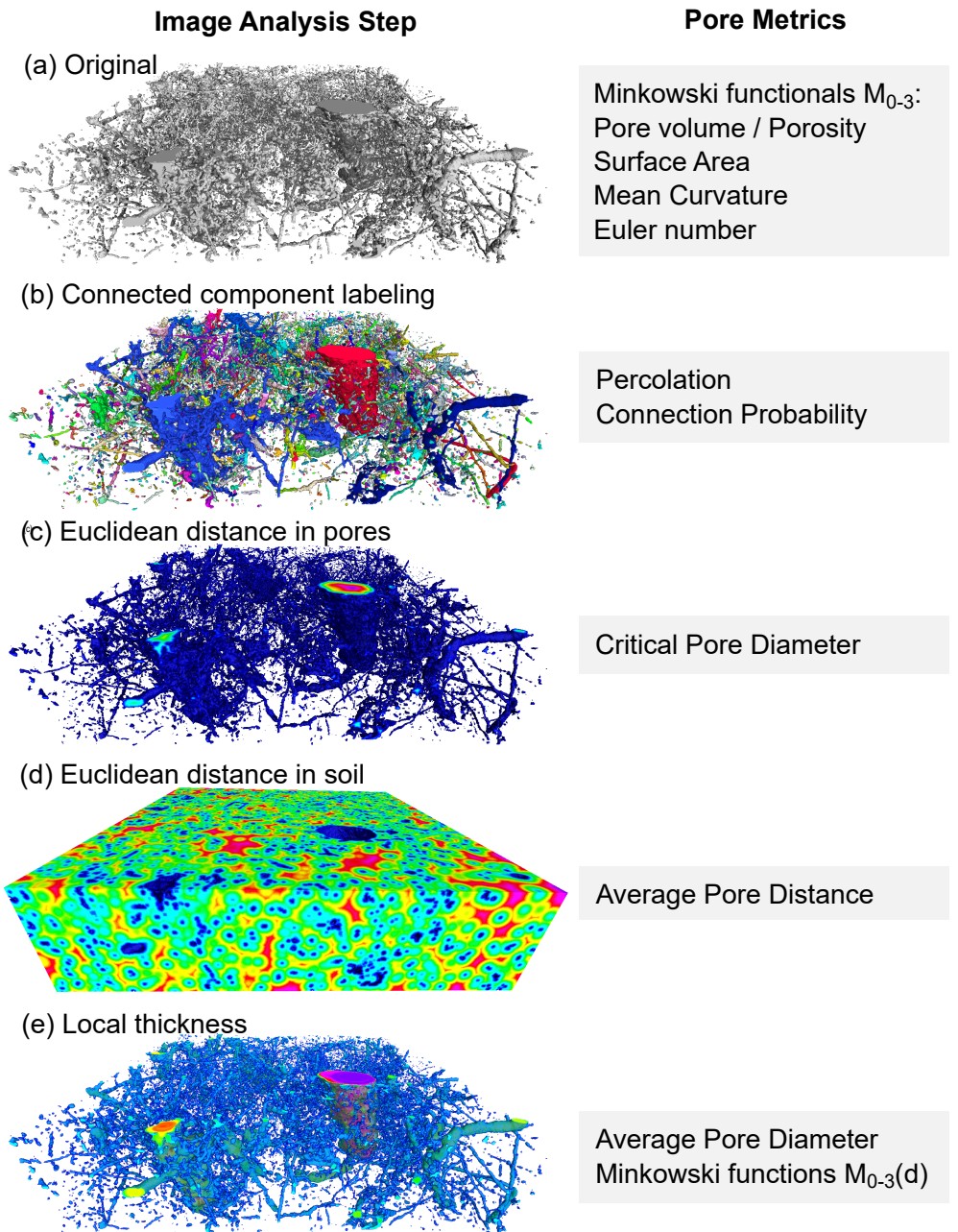

**Image Analysis Step**

(a) Original

(b) Connected component labeling

(c) Euclidean distance in pores

(d) Euclidean distance in soil

(e) Local thickness

**Pore Metrics**

Minkowski functionals $M_{0-3}$:
Pore volume / Porosity
Surface Area
Mean Curvature
Euler number

Percolation
Connection Probability

Critical Pore Diameter

Average Pore Distance

Average Pore Diameter
Minkowski functions $M_{0-3}(d)$

| property | symbol | unit | remarks |
|---|---|---|---|
| ROIvolume | $V_{ROI}$ | [mm$^3$] | entire volume of the structure image or volume of non-zero voxels in the ROI mask |
| porosity | $V_v$ | [-] | visible porosity, equals $m_0$ |
| surface density | $S_v$ | [1/mm] | equals $m_1$ |
| mean curvature density | $\bar{C}_v$ | [1/mm$^2$] | equals $m_2$ |
| Euler characteristic density | $\chi_v$ | [1/mm$^3$] | equals $m_3/4\pi$ |
| percolation | $p$ | [-] | boolean property (0,1) |
| connection probability | $\Gamma$ | [-] | ratio between 0 and 1 |
| critical pore diameter | $d_{cr}$ | [mm] | pore diameter at which percolation is lost |
| contact distance histogram | $h(c)$ | [-] | frequency distribution of pore distances in the solid phase |
| average pore distance | $\bar{c}$ | [mm] | derived from $h(c)$ |
| pore size distribution | $V_v(d)$ | [-] | porosity density for pore diameters $> d$ |
| average pore diameter | $\bar{d}$ | [mm] | derived from $V_v(d)$ |
| surface density distribution | $S_v(d)$ | [1/mm] | surface area density for pore diameters $> d$ |
| mean curvature distribution | $\bar{C}_v(d)$ | [1/mm$^2$] | mean curvature density for pore diameters $> d$ |
| connectivity function | $\chi_v(d)$ | [1/mm$^3$] | Euler characteristic density for pore diameters $> d$ |

**Table 2.** Summary of image analysis results

interpreted as a connectivity metric that turns negative when the number of connections exceeds the number of isolated objects and vice versa.

Minkowksi functionals $M_i$ are extensive properties (meaning, they change their value with the size of the image) calculated for the analyzed volume ($V_{ROI}$). We transform these quantities to densities, $m_i = M_i/V_{ROI}$, to account for the fact that the volume of different datasets can be very different. Any metric derived from $m_{0-3}$ are indicated by the subscript $v$, e.g. $\chi_v$, as these are intensive properties (indifferent to image size).

### 2.2.2 Cluster analysis

As a next step the binary image is separated into individual pore clusters (Fig. 1b) with the Connected Components Labeling method in MorphoLibJ (Legland et al., 2016). Two metrics are retrieved from this image: (a) Percolation is determined as a Boolean property depending on whether at least one pore cluster is present that connects the top and the bottom of the image. (b) The connection probability, also denoted as $\Gamma$ indicator, is retrieved from the second moment of the cluster size distribution,

$$\Gamma = \frac{1}{N_n^2} \sum_{i=1}^{N_l} n_i^2 \tag{2}$$

where each pore cluster has a label $l_i$ and a size $n_i$, expressed as a number of voxels. $N_l$ is the total number of pore clusters and $N_n$ is the total number of pore voxels. $\Gamma$ is one when the entire pore space is connected in one big pore cluster, whereas $\Gamma \to 0$ when the pore space is very fragmented.

### 2.2.3 Distance transforms

The next steps involve an Euclidean distance transform of the pore space (Fig. 1c) and the soil matrix (Fig. 1d). This transform determines the shortest distance to the pore surface of each voxel in the foreground and background, respectively. The critical pore diameter $d_{cr}$ is determined by segmenting the transformed pore space at each distance step to check at which pore diameter percolation breaks down by using connected components labeling. The distance transform of the background is used to compute the contact distance distribution, i.e. the histogram of pore distances within the solid phase $h(c)$, and derive the average pore distance $\bar{c}$ from it.

### 2.2.4 Local thickness

The local pore diameters within the pore space are retrieved with the maximum inscribed sphere method which is called Local Thickness in Fiji. An average pore diameter $\bar{d}$ is calculated based on the histogram of the Local Thickness transform. This transform leads to a pore size distribution $V_v(d)$ which can be related to the water distribution as a function of the capillary pressure (i.e. retention characteristic) assuming spherical interfaces between water and air (Vogel et al., 2010). A similar measure is the medial axes transform where the local pore size is projected onto the skeleton of the pore space. This leads to a different pore size distribution since the volume fractions are obtained from their length along the skeleton. Moreover, this medial axis transform, which is called Skeletonize in Fiji, is very time consuming and therefore discarded here.

In addition, the Minkowski densities $m_{0-3}$ are calculated as a function of pore diameter $m_{0-3}(d)$. This results in the cumulative pore size distribution $V_v(d) = m_0(d)$, the distribution of surface density $S_v(d) = m_1(d)$, the distribution of mean curvature density $C_v(d) = m_2(d)$ and the distribution of the Euler number density $\chi_v(d) = m_3/4\pi(d)$.

## 2.3 Data visualization

The data visualization of both the meta information and the results of the image analysis is implemented with Dash (https://plot.ly/dash), an open source library based on a Python framework for building interactive web applications. It builds on various other packages, such as Pandas and Numpy for data import and transformation as well as Plotly for visualization.

An example of the graphical output with Dash is shown in Fig. 2. It is split into a left frame containing drop-down menus and sliders to create queries for selected data sets based on meta information, e.g. bulk densities larger than x, soil type y, etc. The right frame displays quantitative information in a scatter plot. The assignment of numeric properties to the x- and y-axis can be selected by the user, e.g. $V_v$ and $\Gamma$, respectively, and different colors are assigned to different entities of a query, e.g. different soil types. The ordinate values are also summarized with averages in an additional bar chart. Finally, important meta information like geographical coordinates and texture are displayed for all data sets of the query in interactive maps and ternary

**Figure 2.** Visualization of the Gamma connectivity as a function of porosity for all data sets. The plotted metrics can be selected under the "Graphic" tab. A subset of data can be plotted by using the filter option to select specific site characteristics. Hovering over the single data points provides more information on the soil type and a direct link to the complete metadata of the selected sample.

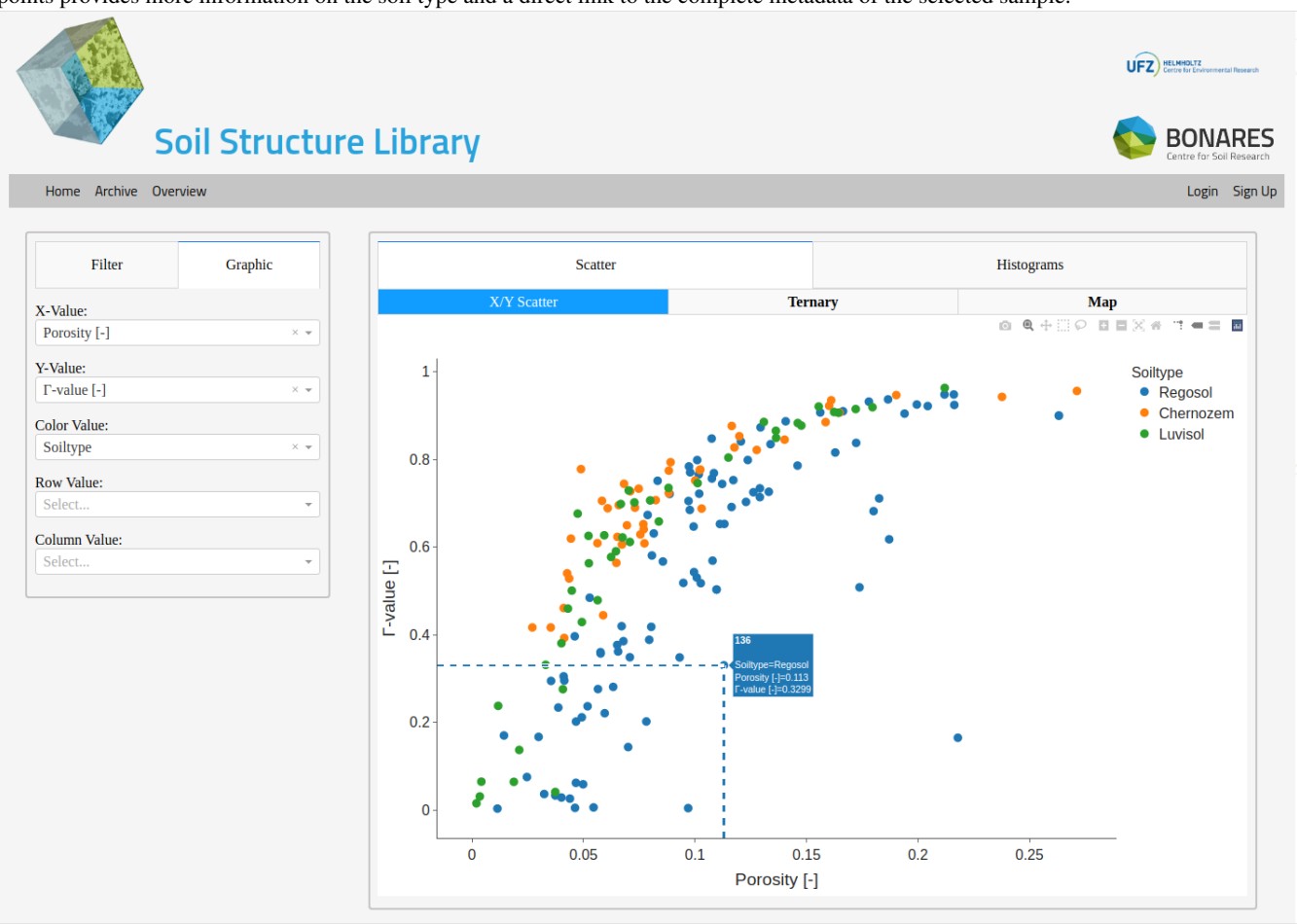

diagrams, respectively. Clicking on individual data points opens the data sheet containing all meta information (Table 1) and image analysis results (Table 2) of that data set. Finally, all interactive graphs can be saved as png images and all underlying data can be exported as csv tables.

## 3 Mining the soil structure library

The soil structure library can be harnessed in various ways. First, it is a data repository that can be used by scientists to upload their segmented X-ray CT data and make it available to the public. This data availability is becoming more important as, for good reasons, an increasing number of scientific journals have introduced a stricter policy in this respect.

Second, segmented X-ray CT data for a large number of different soils is a valuable source of realistic scenarios which can be used for development and testing of three-dimensional, image-based modelling approaches. Examples for image-based modelling could be water flow and matter transport by convection and dispersion (Blunt et al., 2013; Daly et al., 2015), matter turnover by reaction and diffusion (Pot et al., 2022; Falconer et al., 2015; Zech et al., 2022) or maintenance of biodiversity by habitat modelling(Pot et al., 2022; Portell et al., 2018). The model results can be put in a broader context by regression analysis with the morphological properties of the pore structure and with the uploaded meta-information that characterize basic soil properties. In the long run the soil structure library can host a similar amount and variety of pore structures in soil like popular repositories such as the digital rocks portal (Prodanovic et al., 2015) offer for pore structures in rocks mainly for the petroleum engineering science community.

Third, the structure library can be a reference for the suitability of soil as habitat for soil organisms. Structural attributes can be linked to biological activity or the abundance of various species (Hallett et al., 2013; Schlüter et al., 2022a). If such relations are found and can be expressed in the metrics provided, the structure library provides the data base to identify soil types or soil management practices that are expected to impact the soil biome and its activity in one way or another. This also includes soil processes that emerge from the interplay of pore structure and microbial activity like the formation of anaerobic soil volumes and green house gas formation (Rabot et al., 2015; Kravchenko et al., 2018; Rohe et al., 2021).

Fourth,the structure library can be mined in order to deduce general patterns, relationships or tipping points that may exist among structural properties or between basic soil properties and structural properties. A short example shall suffice here to demonstrate such a data mining approach.

## 4   A case study on connectivity metrics

Several metrics have been implemented that quantify different aspects of pore connectivity. Percolation represents the existence of a continuous path between image borders, i.e. long range connectivity between distant locations in the pore space. The critical pore diameter $d_{cr}$ indicates the size of the smallest pore neck in this path. The Euler characteristic reflects the intrinsic connectivity independent of location or distance. It does not provide any information on the length scale of connections, but about the internal number of connections independent of whether they are percolating or not. It has been conjectured that, under certain conditions of structural homogeneity, the percolation threshold the number of isolated objects and redundant loops are exactly balanced, i.e. $\chi = 0$ (Mecke, 2000; Vogel et al., 2010). The corresponding minimum pore diameter when this balance is reached shall be denoted as $d_{\chi_0}$. The transition in connection probability $\Gamma$ from fully connected to completely fragmented is expected to occur in a similar pore diameter range. It will decrease monotonically when small pores are removed sequentially in a series of increasing minimum pore diameters. Until now, there has been no comprehensive analysis as to (1) whether long-range connectivity and intrinsic connectivity break apart around the same pore diameter and (2) what the remaining pore volume $V_v$ and connection probability $\Gamma$ at these pore diameters ($d_{cr}$ and $d_{\chi_0}$) are. In addition, these connectivity metrics may serve as a fingerprint of the pore structure that is able to distinguish between pore systems that are generated by different processes. Such an approach will be demonstrated here for a selection of soil samples with identical resolution ($20\mu$m) and

180    similar soil properties (texture, SOM content, climate, etc.) but managed as long-term conventional tillage (CT) or no-tillage (NT). Samples for both treatments originate from different locations (Schlüter et al., 2018; Lucas et al., 2019). All samples without a percolating pore cluster were sorted out beforehand and all isolated pores in the original pore network were removed in order to ensure that the analyzed pore space is well connected and $\chi$ of the entire percolating cluster is negative. The complete data set comprises 104 samples (CT:34, NT: 70).

**Figure 3.** Connectivity metrics to analyze characteristic pore morphologies imposed by different tillage treatments: (a) Euler number density as a function of minimum pore diameter for all investigated samples. The vertical lines mark the median pore diameters where intrinsic connectivity (dashed lines) and long-range connectivity (solid lines) is lost. (b) Relationship between $d_{cr}$ and $d_{\chi 0}$ for individual samples and treatment statistics. (c) Cumulative pore size distribution and the pore volume where $d_{cr}$ and $d_{\chi 0}$ is reached either for all investigated samples or as treatment statistics. (d) Connection probability $\Gamma$ at $d_{cr}$ and $d_{\chi 0}$ for both tillage treatments. Different letters represent significant differences (p<0.05, Wilcoxon rank sum test).

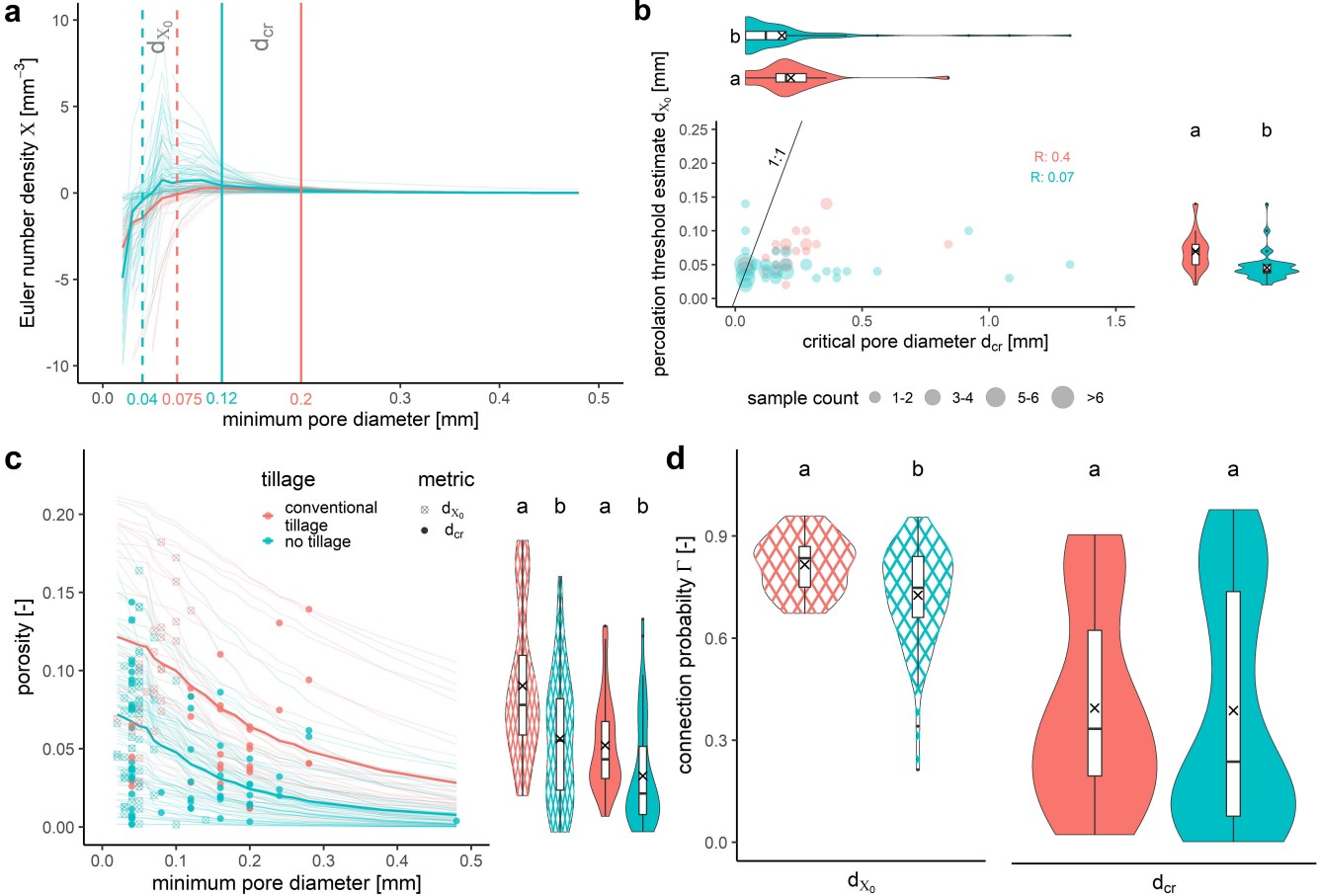

It turns out that $d_{\chi_0}$ is much smaller than $d_{cr}$ irrespective of tillage treatment (Fig. 3a-b). This is because $d_{\chi_0}$ indicates, at which pore diameter the removal of pore necks due to morphological openings has created as many isolated pores as remaining redundant connections, whereas $d_{cr}$ indicates when the last pore object that still sustains vertical percolation is lost towards the end of this succession of pore removal steps. Both, $d_{\chi_0}$ and $d_{cr}$ are significantly higher in pore structures produced by plowing (3b). That is, the fragmentation of the soil through mechanical disturbance forms a network of large macropores with higher connectivity. There, are occasional outliers for $d_{cr}$ in the NT treatment that represent samples with at least one large continuous biopore from top to bottom that is not refilled by casts or soil fragments. In fact, these outliers even lead to similar $d_{cr}$ averages (crosses in violin plots) despite significantly different populations in terms of rank metrics(Fig. 3b).

The very different pore morphology with and without plowing also manifests itself in the remaining porosity at the minimum pore diameter where $d_{\chi_0}$ and $d_{cr}$ occur (3c). The pore structure in undisturbed soil (NT) is dominated by cracks and biopores with elongated, planar or cylindrical shapes. That is, they stretch across long distances with rather small volumes. The pore structure after plowing, in turn, is dominated by more bulky and isotropic packing pores with a lower spatial extent per volume. This is why the pore structure in NT soils needs significantly less porosity to sustain both $d_{\chi_0}$ and $d_{cr}$.

As a consequence, the connection probability of the remaining porosity at $d_{\chi_0}$ is larger in CT soils (3d), because more of the NT soils have already reached a critical macroporosity range around 0.05-0.1 at which $\Gamma$ decreases sharply. At $d_{cr}$ both NT and CT pore structures are in this critical macroporosity range. As a result, there is a huge variability in $\Gamma$ with fluctuations across the entire possible range. In addition, both treatments exhibit a distinct bimodal distribution of $\Gamma(d_{cr})$ enforced by the non-linear relationship between visible porosity and connection probability.

Finally, it has been conjectured before (Schlüter et al., 2011; Lucas et al., 2019) that biopores produced by fine roots with typical diameters of 0.1-0.2 mm are the main contributor to pore connectivity in no-till samples. This is confirmed with this case study by the fact that (i) $d_{cr}$ falls into this root diameter range and (ii) $\Gamma$ also reaches 0.5 around $d_{cr}$.

## 5  Conclusions

With the Soil Structure Library we have established a free platform for sharing segmented X-ray CT images of pore images among the soil science community. The library can be used as a conventional data repository to provide access to 3D large image data, which is a service that has not been available until now, but is becoming more important with updated data policies of many journals. Likewise, the soil structure library is a rich source of realistic 3-dimensional pore structures for image-based modelling on a large range of image resolutions and domain sizes. Access to such image data is appealing especially to scientists with no or limited access to imaging facilities. In a similar vein, the soil structure library offers free, standardized and reproducible soil structure analysis to users who lack the computing infrastructure or expertise for pore structure analysis. The full potential of the soil structure library enfolds, however, when harnessed for data mining and regression analysis with complementary meta-information in order to better understand the relationship between soil structure and soil functions.

*Author contributions.* Ulrich Weller set up the technical framework of the structure library, wrote and edited, Lukas Albrecht worked on the graphics and wrote, Steffen Schlüter set up the calculation routine and wrote, Hans-Jörg Vogel accompanied the scientific background and wrote

*Competing interests.* There are no competing interests

220 *Acknowledgements.* We thank John Koestel and David Legland for adaptions to SoilJ and MorpholibJ to streamline them with our ImageJ script. This work was funded by the German Federal Ministry of Education and Research (BMBF) in the framework of the funding measure Soil as a Sustainable Resource for the Bioeconomy – BonaRes, project BonaRes (Module B): BonaRes Centre for Soil Research (grant 031B0511).

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
