# Peer review of "An Open Soil Structure Library based on X-ray CT data"

_SOIL, 2021_

## Author Response (AR1)

Reviewer 1

The idea of an open soil structure library based on X-ray CT data is great. The authors have an extensive expertise in the subject. The manuscript is well organized. The case study is relevant. There are two aspects where I can see an improvment of the present manuscript. First, even if the authors have been working with this subject for many years, they are not the first and not the only ones. I suggest to reduce the selfcitation to the absolute necessary (). For example, L18, L24 (can't find Schlüter et al., 2019 in the reference list by the way), L26, and L45, paperspublished  by others than the authors should be cited for these topics, and it doesn't matter if these manuscripts are older. Second, as mentioned by the authors, segmentation is crucial for the resulting binary image of the soil structure. Other sensitive procedures are image acquisition (type and setting of the scanner) and filtering (physical and numerical). I suggest to request metadata concerning the acquisition, the filtering, and the segmentation for each uploaded dataset in the way as soil and site properties.

Reply:
Many thanks for the kind review. We are happy to follow the advise given. The requested metadata on segmentation helps with the interpretation and is added to the library.  The reviewer has a good point that we have to broaden our literature cited and we are replacing the literature as recommended. (Schlüter et al. 2019 is given in line 263 by the way)

We have replaced and added the literature, papers of Larsbo et al, Tracy et al., Kravchenko et al, and Juarez were added to the reference list and self citations were diminuished.

The table of metadata was increased by the information on segmentation.

Reviewer 2:

In this paper an online accessible library of soil structure analyses to be used with soil 3D X-ray CT data is introduced. Over the past few decades use of CT to elucidate soil structure has been booming and the authors rightfully point out that standardization of methods used for investigating such soil 3D data is much needed. Very uniquely a system has been set up into which any user could upload a binary soil CT data set, which is then subjected to a set of analyses on a server and results are sent back to the user. A very commendable initiative on its own to get analyses standardized, but moreover also the launch of the online library of soil-CT data holds strong added value to address to date under lighted research questions on soil structure.

This paper is atypical and reads more like a manual at times, yet in my view it forms a valuable contribution to literature as now the Soil Structure Library will be formally introduced to the soil science audience. The text is very well written and accessible in spite of the rather complex and abstract matter dealt with. One comment though, it would have been interesting to see more contemplation on how this Library might help scientists looking into the relation between soil structure and biochemical processes (OM breakdown, anaerobic soil microbial activity, etc.). Ideally the authors elaborate a bit more on such uses in 3. and in doing so may surely make a case for the Library to the currently wide community of soil biologists & biochemists.

Reply: we have added literature for pointing out the use of 3D images in modelling and have added a paragraph on biological activity.

„Third, the structure library can be a reference for the suitability of soil as habitat for soil organisms. Structural attributes can be linked to biological activity or the abundance of various species \citep{hallett+2013,SCHLUTER+2022}. If such relations are found and can be expressed in the metrics provided, the structure library provides the data base to identify soil types or soil management practices that are expected to impact the soil biome and its activity in one way or another. This also includes soil processes that emerge from the interplay of pore structure and microbial activity like the formation of anaerobic soil volumes and green house gas formation \citep{Rabot+2015,Kravchenko+2018,Rohe+2021}.“

Specific comment:

P3 Would it not be meaningful to also by default inquire whether a soil core was repacked or collected undisturbed?

We have added this information to the metadata.

P4 L71 So does this then mean that ROIs cannot be confounded to a selection of horizontal slices? Not entirely clear

We have reformulated the passage and it now says „The ROI image can be uploaded in form of a binary image (tiff or mhd/raw) with identical 3D dimensions as the uploaded image, or in form of a selection (roi format) to be created and exported in ImageJ. This selection is a two-dimensional image and will be applied to all slices equally (e.g. a circular selection will result in a cylindrical mask). All files have to be uploaded in one compressed zip-folder.“

P6 L99 What is intended by 'intensive properties' this requires further explanation

We have added an explanation on extensive and intensive properties

„Minkowksi functionals $M_{i}$ are extensive properties (meaning, they change their value with the size of the image) calculated for the analyzed volume ($V_{ROI}$).“

„Any metric derived from $m_{0-3}$ are indicated by the subscript $v$, e.g. $\chi_v$, as these are intensive properties (indifferent to image size).“

P6 L 106, so ni is expressed in number of voxels then?

We have added this. „each pore cluster has a label $l_i$ and a size $n_i$, expressed as a number of voxels.“

L120-123 the 'medial axes transform' is not offered, and so its brief mention here is only confusing -> suggest to omit these two sentences.

We are often asked why we do not use the medkial axis transform and therefore want an explanation here why we preferred the local thickness transform.

L131-132 this mention of benefits for a certain public domain is redundant for this paper; in fact the power of Dash becomes soon evident when reading through the following lines.

We have deleted this sentence.

In 2.2.4 it is not clear how to interpret several of the pore metrics as a function of pore diameter m0-3 (d): does such a metric represent all pores with local pore diameter <d, >d or with d in between to pore size classes -> the latter would be of most interest to users. Please clarify

The number is given in local pore diameter > d, from this distribution it is easy to calculate the desired interpretation. We would like to keep it this way, as this is the way it is directly defined and the class wise distribution is derived from that number.

From the conclusions section it would seem that the actual 3D datasets are also accessible via the Soil Structure Library. Not clear if that is really the case? Please clarify in the manuscript.

We have added in the abstrqact and the text that this is the case and that the images are accessible.

Technical corrections:

1st line of the abstract: 'and' between 'physical' and 'biochemical'

P1L21 be consistent x-ray or X-ray

Table 2 needs to be positioned further on after it has been invoked in 2.2

We have tried to change this, but the placement of tables and figures is done by the final formatting by the editorial preocess.

P6 Reference is made to Fig.1 a-d yet such alphabetical numbering was not included in the actual Figure; please add.

P7 L 114 perhaps do repeat the symbol for these soil pore metrics here in the text as well, e.g. for average pore distance.

L155 perhaps better petroleum engineering "science" community?

Caption Fig 3 last sentence : 'represent' not 'represents'

L206 'scientists'

All technical corrections have been applied.